# The Mla pathway in *Acinetobacter baumannii* has no demonstrable role in anterograde lipid transport

Matthew J Powers[1,2], Brent W Simpson[1], M Stephen Trent[1,2]*

[1]Department of Infectious Diseases, College of Veterinary Medicine, University of Georgia, Athens, United States; [2]Department of Microbiology, College of Arts and Sciences, University of Georgia, Athens, United States

**Abstract** The asymmetric outer membrane (OM) of Gram-negative bacteria functions as a selective permeability barrier to the environment. Perturbations to OM lipid asymmetry sensitize the cell to antibiotics. As such, mechanisms involved in lipid asymmetry are fundamental to our understanding of OM lipid homeostasis. One such mechanism, the Maintenance of lipid asymmetry (Mla) pathway has been proposed to extract mislocalized glycerophospholipids from the outer leaflet of the OM and return them to the inner membrane (IM). Work on this pathway in *Acinetobacter baumannii* support conflicting models for the directionality of the Mla system being retrograde (OM to IM) or anterograde (IM to OM). Here, we show conclusively that *A. baumannii mla* mutants exhibit no defects in anterograde transport. Furthermore, we identify an allele of the GTPase *obgE* that is synthetically sick in the absence of Mla; providing another link between cell envelope homeostasis and stringent response.

*For correspondence:
strent@uga.edu

Competing interests: The authors declare that no competing interests exist.

## Introduction

The outer membrane (OM) of the Gram-negative cell envelope is the defining characteristic and a fundamental organelle of these organisms (*Henderson et al., 2016*). The OM, which separates the aqueous periplasmic space from the environment, serves as a critical, selective permeability barrier that prevents the entry of noxious compounds (*Simpson and Trent, 2019*). How the OM fulfills this role is two-fold and is predicated on its asymmetric lipid composition (*Nikaido, 2003*). While the inner leaflet of the OM is composed of the standard glycerophospholipids (GPLs), the outer leaflet is predominantly lipopolysaccharide (LPS) or lipooligosaccharide (LOS) (*Funahara and Nikaido, 1980*; *Schindler and Osborn, 1979*). LPS or LOS differ by the presence or absence of O-antigen sugar repeats respectively, but both molecules are anchored in the OM by lipid A, a unique glycolipid, that is remarkably conserved amongst Gram-negative bacteria (*Whitfield and Trent, 2014*). Lipid A is a *bis*-phosphorylated glucosamine disaccharide that can be anywhere from tetra- to hepta-acylated depending on genera (*Whitfield and Trent, 2014*). This increased hydrophobicity is the first reason for the OM's functionality as a barrier, as it strongly excludes the passive entry of hydrophilic compounds (*Nikaido, 2003*). Secondly, negatively charged phosphates on lipid A and covalently attached sugars that make up the core of LPS/LOS increases the net negative charge of the outer membrane. This facilitates efficient cross-bridging with divalent cations in the environment and promotes strong lateral interactions between LPS/LOS molecules on the surface (*Schindler and Osborn, 1979*; *Murata et al., 2007*). These tightly packed LPS/LOS molecules, particularly the core and O-antigen sugars, serve as physical impediments to the accessibility of hydrophobic compounds to the lipid bilayer of the OM (*Funahara and Nikaido, 1980*; *Schindler and Osborn, 1979*).

When taking these factors into consideration, it is unsurprising that the OM has remained one of the largest hurdles in an increasingly desperate race for antimicrobial discovery (*Lehman and*

*Grabowicz, 2019*; *Luepke and Mohr, 2017*). A large majority of antibiotic classes are ineffective on Gram-negative bacteria due to either the inability to diffuse across the membrane (*Nikaido, 1994*; *Vergalli et al., 2020*) or rapid efflux out of the cytoplasmic space (*Du et al., 2018*). As such, the intrinsic mechanisms behind the biosynthesis and maintenance of the OM has been a high priority research focus for decades. Through the work of many, the field has accelerated significantly such that we know many of the players involved in these processes with varying degrees of mechanistic detail behind their function. The Lipopolysaccharide transport (Lpt) pathway forms a transmembrane bridge to transport LPS/LOS to the OM (*Okuda et al., 2016*). The Localization of lipoprotein (Lol) system transports lipoproteins from the IM to the OM (*Grabowicz, 2019*). The β-barrel assembly machinery (Bam) facilitates folding and insertion of outer membrane proteins (OMPs) (*Malinverni and Silhavy, 2011*). However, one of the more elusive systems have been those that transport GPLs between the IM and OM. From a first principle, we know that this process must occur in an anterograde direction (IM to OM). Additionally, previous work done by Mary Jane Osborn's group highlighted the ability for bacterial cells to rapidly and efficiently transport GPLs in the retrograde direction (OM to IM) (*Jones and Osborn, 1977*). There is evidence that deletion of the Tol-Pal system negatively impacts bulk retrograde GPL transport, although whether this is direct or indirectly mediated by Tol-Pal remains to be determined (*Shrivastava et al., 2017*). Outside of this example, we lack significant knowledge on how GPL transport occurs in either direction, with most of our current understanding limited to a single pathway: *M*aintenance of *l*ipid *a*symmetry (Mla) (*Powers and Trent, 2019*).

The Mla pathway was discovered in 2009, where it was found that mutants of *E. coli* lacking this pathway were susceptible to combinatorial exposure to SDS/EDTA (*Malinverni and Silhavy, 2009*). Susceptibility to detergent + chelator combinations indicate perturbed OM asymmetry, as *E. coli* is typically highly resistant due to the lateral LPS interactions (*Schindler and Osborn, 1979*). Spontaneous suppressors that could alleviate sensitivity all achieved the same result – increased expression of *pldA* (*Malinverni and Silhavy, 2009*). PldA is an outer membrane phospholipase which, demonstrated through extensive crystallographic and biochemical analyses, has an active site that is exclusively exposed to the outer leaflet of the OM (*Brok et al., 1996*; *Scandella and Kornberg, 1971*; *Snijder et al., 1999*). As such, PldA enzymatically degrades GPLs that are mislocalized to the outer leaflet of the OM. Through this suppression analysis, it was reasonably hypothesized that Mla and PldA have overlapping functionalities albeit through distinct mechanisms (*Malinverni and Silhavy, 2009*).

How Mla mediates extraction and removal of mislocalized GPLs, and whether this is ultimately its function, remains a point of controversy. The intact, functional system is comprised of 6 proteins (*Powers and Trent, 2019*). MlaA is an alpha-helical outer membrane lipoprotein (*Abellón-Ruiz et al., 2017*). MlaC is a soluble, periplasmic protein (*Huang et al., 2016*). MlaD is anchored in the inner membrane with a lipid-binding MCE domain in the periplasmic space (*Ekiert et al., 2017*; *Thong et al., 2016*). It is coupled in a tight complex with an inner membrane permease, MlaE, a cytoplasmic ATPase, MlaF, and a cytoplasmic accessory protein MlaB (*Ekiert et al., 2017*; *Thong et al., 2016*). Together, this system comprises all components that would be theoretically necessary to mediate lipid transport. Structural analyses and subsequent genetic data further bolstered the hypothesis that the Mla pathway functions in the initially proposed retrograde manner (*Abellón-Ruiz et al., 2017*; *Baarda et al., 2019*; *Chong et al., 2015*; *Sutterlin et al., 2016*). This included a study done by our group in *Acinetobacter baumannii,* where we used experimental evolution to assess fitness factors in the total absence of outer membrane asymmetry (*Powers and Trent, 2018a*). For space reasons, we defer to our recently published perspective that consolidates and analyzes the wealth of Mla data in the literature (*Powers and Trent, 2019*).

*A. baumannii* is relatively unique in that it can survive in the absence of LOS, which is typically essential in Gram-negative organisms (*Powers and Trent, 2018b*). In this case, the lipid composition of its OM is exclusively a symmetric GPL bilayer (*Powers and Trent, 2018a*). What we found overwhelmingly was that when given the opportunity, LOS-deficient populations across multiple strains of *A. baumannii* rapidly inactivated genes in the Mla pathway and occasionally *pldA* (*Powers and Trent, 2018a*), a finding that was independently corroborated by the Kahne group (*Nagy et al., 2019*). Our interpretation of these data strongly supported that of the retrograde model for transport.

Early in 2019, Kamischke et al. argued that the Mla pathway in *A. baumannii* functions instead as an anterograde GPL transporter (*Kamischke et al., 2019*). This was surprising to us as it clashed quite directly with our previous findings in the same organism. We were curious as to how this discrepancy arose and sought to reassess the validity of our initial findings in light of this new study. To this end, we requested the Δ*mlaF* mutant and its isogenic parent used in Kamischke et al. to determine if strain differences may have resulted in incompatible interpretations.

In comparing strains, a number of differences became readily apparent, including a strong growth defect in the Δ*mlaF* mutant from the Kamischke et al. study. As a point of clarity, strains derived in the Trent Lab will be annotated with 'UGA' and strains from Kamischke et al. will be annotated as 'UW' (the home institution of the corresponding author). This growth defect was surprising as as no *mla* mutants in *A. baumannii* (*Powers and Trent, 2018a*), *Burkholderia cepacia* (*Bernier et al., 2018*), *Neisseria gonorrhoeae* (*Baarda et al., 2019*), *Haemophilus influenzae* (*Fernández-Calvet et al., 2018*), or *E. coli* (*Ekiert et al., 2017*; *Chong et al., 2015*) have been shown to have any growth defect compared to the isogenic parent. We link the growth defect of the UW Δ*mlaF* to a mutation present in *obgE*. ObgE is a GTPase involved in the stringent response (*Persky et al., 2009*). The stringent response in bacteria revolves around the accumulation of the GTP-derived nucleotide alarmones (p)ppGpp (*Zhu et al., 2019*). Accumulation of this alarmone is typically triggered by nutritional stress and remodels the transcriptome to allow for tolerance and growth under limited conditions (*Zhu et al., 2019*). Synthesis of this secondary messenger is mediated by two proteins, RelA and SpoT (*Pérez-Varela et al., 2020*). While RelA is the primary synthase enzyme, SpoT can both synthesize and hydrolyze ppGpp (*Hernandez and Bremer, 1991*). There is additional evidence suggesting that ObgE and SpoT co-purify in a potentially physiologically relevant manner (*Wout et al., 2004*) and that ObgE plays a role in stringent response nucleotide homeostasis (*Persky et al., 2009*). As such, we hypothesized that the allele of *obgE* in the UW strain, termed here as *obgE\**, plays an aberrant role in the stringent response, which is further exacerbated in the absence of a functional Mla system.

Lastly, we devised an assay to detect defects in anterograde transport that monitors outer membrane vesicle (OMV) formation with a pulse-chase in lieu of the membrane separations and LC-MS/MS assay used by Kamischke et al. At this time, Kamischke et al. is the only publication reporting the successful separation of *A. baumannii* inner and outer membrane fractions, which we ultimately found to be unreproducible (*Kamischke et al., 2019*). Both our lab and the Dalebroux lab have found the methodology to be insufficient as reported here and in JOVE (*Cian et al., 2020*). As we were unable to reliably separate membranes of *A. baumannii,* we utilized the OMV pulse-chase to circumvent these caveats. Using this assay, we found that no *mla* mutants, regardless of lab origin, exhibited a decrease in anterograde transport.

In the following manuscript, we provide evidence on two counts. The first is that *mla*-null mutants in *A. baumannii* are *not defective in anterograde transport*. The second explores in mechanistic detail an allele of *obgE* that we argue is responsible for the phenotypic abnormalities of the UW Δ*mlaF* strain. We are confident that these data, in conjunction with the wealth of previously published data on Mla, strongly support the prevailing model that Mla mediates retrograde transport of mislocalized GPLs in Gram-negative bacteria.

## Results

### Comparison of *mlaF* mutants

In Kamischke et al., the authors argue that the Mla pathway mediates anterograde GPL transport in *A. baumannii*. As this ran counter to published data, including work done by our group in the same organism, we wanted to explore strain differences that could result in such different conclusions. The UW Lab generously provided their *A. baumannii* Δ*mlaF* mutant and its isogenic parent. Immediately it was clear something was different between the strains. While the UGA Δ*mlaF* mutant grows identically to its isogenic parent (red and black triangles, respectively), the UW Δ*mlaF* strain exhibited a significant growth defect (green and orange circles, respectively) (*Figure 1A*). Quantification by colony forming units from hours 1 to 4 confirmed that, despite its unexpected growth defect, increases in optical density during exponential phase were due to growth (*Figure 1B*). Furthermore, the UW Δ*mlaF* strain appeared morphologically distinct in both size and color from its parent when

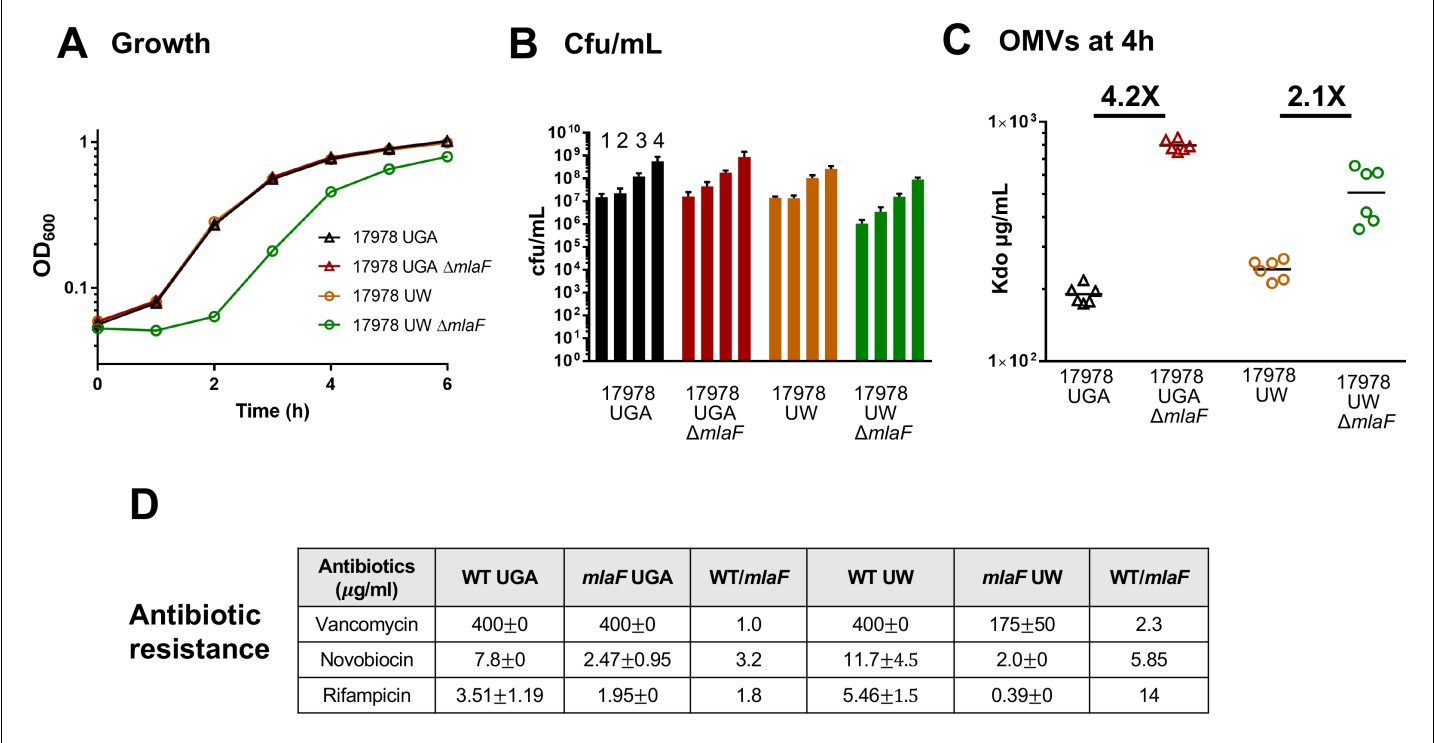

**Figure 1.** Δ*mlaF* mutant from the UW group exhibits significant growth defects. (A) Growth curves of WT and Δ*mlaF* deletion mutants from the UGA lab (triangles) and UW lab (circles). Standard deviation error bars, if smaller than symbols, are not shown. (B) Colony-forming units (cfu) of cultures at hours 1–4. (C) Outer membrane vesicle (OMV) quantification of WT and Δ*mlaF* deletion mutants harvested at 4 hr and quantified by measuring 3-deoxy-D-*manno*-oct-2-ulosonic acid (Kdo) levels using the Purpald assay. Growth curves were performed in biological triplicate. OMV assays were performed in biological duplicate and quantified in technical triplicates. (D) MICs for vancomycin, novobiocin, and rifampicin for both UGA and UW strains. MICs were calculated as described in the Materials and methods.

The online version of this article includes the following figure supplement(s) for figure 1:

**Figure supplement 1.** UW Δ*mlaF* has a distinct colony morphology.
**Figure supplement 2.** OMVs collected from stationary phase cultures at 24 hr.
**Figure supplement 3.** *obgE\** alleles are selected against in the absence of Mla.

grown on agar plates (*Figure 1—figure supplement 1*). As mentioned previously, no published *mla* mutants exhibit any growth defects across genera.

One of the major phenotypes of *mla* mutants is an increase in production of OMVs (*Roier et al., 2016*). OMVs are a natural byproduct of Gram-negative growth and cell division, resulting in the sloughing of the OM (*Haurat et al., 2015*). While this occurs naturally to varying extents across bacterial genera, mutations that perturb OM homeostasis can alter the rate of vesiculation. To determine whether the UW Δ*mlaF* mutant also had an altered OMV phenotype, we harvested OMVs at two phases of growth; late exponential phase (*Figure 1C*) and stationary phase (*Figure 1—figure supplement 2*). OMVs are then quantified via a colorimetric Kdo assay due to the exclusive incorporation of this sugar into the LPS/LOS of bacterial cells. During exponential phase, both the UGA and UW Δ*mlaF* mutants exhibited ~2–4X increase in OMVs (*Figure 1C*), consistent with published data in other organisms (*Roier et al., 2016*). Strikingly, when we assessed stationary phase cultures for OMVs, there was a major discrepancy in the relative degrees of vesiculation between mutant and WT. Whereas the UGA Δ*mlaF* mutant maintained fold-changes relative to WT within the expected range, the UW Δ*mlaF* mutant had a staggering 23X increase in OMV levels (*Figure 1—figure supplement 2*). These data suggest that the UW Δ*mlaF* strain has a unique stationary phase lysis phenotype which has not been observed in other *mla* mutants.

Having a growth defect in conjunction with a deletion in a cell envelope related gene suggested to us that the permeability barrier may have been uniquely compromised in the UW Δ*mlaF*. To test this, we utilized three antibiotics to probe for OM integrity: vancomycin, novobiocin, and rifampicin.

In the absence of Mla, one would expect an increase in sensitivity to the small, hydrophobic antibiotics novobiocin and rifampicin. Indeed, this was observed for both the UGA and UW Δ*mlaF* mutants albeit to varying extents. When we examined the minimum inhibitory concentration (MIC) for novobiocin, there was a 3.2-fold increase in sensitivity in the UGA Δ*mlaF* strain compared to a 5.85-fold increase in sensitivity of the UW Δ*mlaF* strain (*Figure 1D*). This trend of exacerbated sensitivity in the UW Δ*mlaF* were also seen with rifampicin, with 1.8-fold and 14-fold increases in sensitivity of the UGA and UW Δ*mlaF* respectively (*Figure 1D*). The vancomycin results were particularly distinct. The UGA Δ*mlaF* mutant exhibited no change in MIC, while the UW Δ*mlaF* has a 2.3-fold increase in sensitivity. This severe, unique sensitivity of the UW Δ*mlaF* to vancomycin provided further evidence to us that the UW Δ*mlaF* strain was exhibiting phenotypic abnormalities when compared to other Mla-null strains.

## Mla mutants exhibit no defects in anterograde transport

The critical finding in Kamischke *et. al.* was a rather significant ~50% decrease in GPL content in the OM of Δ*mla* mutants (*Kamischke et al., 2019*). This was determined using an elegantly designed pulse-chase assay, wherein cells were pulsed with $^{14}$C-acetate, total membranes were separated into IM and OM fractions, and GPL content was measured quantitatively using LC-MS/MS (*Kamischke et al., 2019*). Despite an exhaustive approach, attempts in our laboratory to achieve membrane separations in *A. baumannii* were met with no success (*Figure 2—figure supplement 1*). We noticed significant deviations in the method described in Kamischke et al. from that of the gold standard methodology published by Mary Jane Osborn. Initially, we assumed % sucrose solutions in Kamischke et al. were derived using the standard weight by weight (w/w) calculation (*Kamischke et al., 2019*; *Dalebroux et al., 2015*). This gradient resulted in minimal to no separation of any membranes, including *E. coli*. After obtaining a detailed protocol from Kamischke et al., we realized solutions were calculated as weight by volume (w/v), a difference with significant repercussions to the overall density profile of the gradient. However, even after adjusting the sucrose solutions with this protocol, we were unable to separate *A. baumannii* membranes robustly. For *E. coli*, both methodologies resulted in distinct membrane peaks as indicated by NADH oxidase activity (IM marker) and the presence of LPS (OM marker) (*Figure 2—figure supplement 1A & C*). In contrast, *A. baumannii* membranes failed to separate sufficiently even if they appeared to do so visually (*Figure 2—figure supplement 1B & D*). Using the Osborn sucrose gradient protocol, *A. baumannii* membranes fail to separate at all – collapsing into a single peak (*Figure 2—figure supplement 1B*). When we applied the protocol provided by the UW Lab, we found the results to be of intermediate quality. Visually, we observed multiple bands in the gradient (*Figure 2—figure supplement 1D*). Unfortunately, the OM markers were distributed across almost the entire gradient (*Figure 2—figure supplement 1D*). This highlights the importance of assessing markers across the entire gradient, as this separation technique is imperfect even with the most amenable species (e.g. *E. coli*). Evaluating the entire gradient for markers ensures capturing the full degree, or lack thereof, of separation. Kamischke et al. did not report marker analysis across the gradient, but instead only for ambiguous fractions designated IM and OM. Without knowing what fractions were consolidated, it is difficult to determine the quality of the downstream quantitative LC-MS/MS analysis (*Kamischke et al., 2019*). Independently, a recent methods paper published during the review of this work demonstrated that the gradient utilized by Kamischke et al. was insufficient to separate *A. baumannii* membranes, further validating our results presented here (*Cian et al., 2020*).

Because of inconsistencies in membrane separations of *A. baumannii*, we devised a pulse-chase assay that could indirectly assess anterograde transport independent of membrane separations. To do this, we relied on OM vesiculation as a proxy for anterograde transport, as all OMVs contain GPLs that have undergone anterograde transport from the IM to OM. In short, we labeled exponentially growing cells with $^{32}$P$_i$ for an hour followed by an equal length chase with cold potassium phosphate, resulting in a monitoring of de novo lipid synthesis for 1 hr over the course of 2 hr of growth. Importantly, OMV production is not impacted by cell lysis during the time frame of the OMV pulse-chase assay (*Figure 1*). After the chase period, cells were pelleted, and the supernatant containing OMVs was removed and converted into a two-phase Bligh-Dyer. This serves to extract all lipidic material from the growth supernatant, which can then be quantified using liquid scintillation counting (LSC) and thin-layer chromatography (TLC). In the context of this pulse-chase, any labeled lipid must have been synthesized de novo and anterograde transported to the OM prior to blebbing off

into the supernatant (*Figure 2A*). If *mla* mutants are defective in anterograde transport to the levels indicated in Kamischke et al., we would expect to see a 50% reduction in counts relative to WT. Instead, we observed a significant increase in $^{32}$P-labeled lipids in the supernatant in every Δ*mla* mutant tested (*Figure 2B & C*). Both the UGA Δ*mlaF* and UW Δ*mlaF* mutant had *increased* $^{32}$P-labeled lipids using this assay, indicating no defect in anterograde transport (*Figure 2B & C*). TLC-visualization confirmed that LSC counts were being derived from GPLs, further validating the LSC quantification (*Figure 2B*). To remove strain differences as a variable, we constructed deletions of Δ*mlaC* in the UW WT background (UW *mlaC::kan* MJP1 and MJP2). Consistent with previous results, we saw no defects in anterograde transport as evidenced by an increase in $^{32}$P-labeled OMVs (*Figure 2B & C*).

One potential caveat to this approach would be contamination of IM material that could skew interpretation of results. To verify that our assay was directly monitoring OMVs secreted into the supernatant, we assessed cold OMVs collected in an identical fashion as to the radiolabeled pulse-chase assay. These OMVs were then quantified for presence of an OM marker (OmpA) and the enzymatic activity of an IM protein (NADH Oxidase) (*Figure 3*).

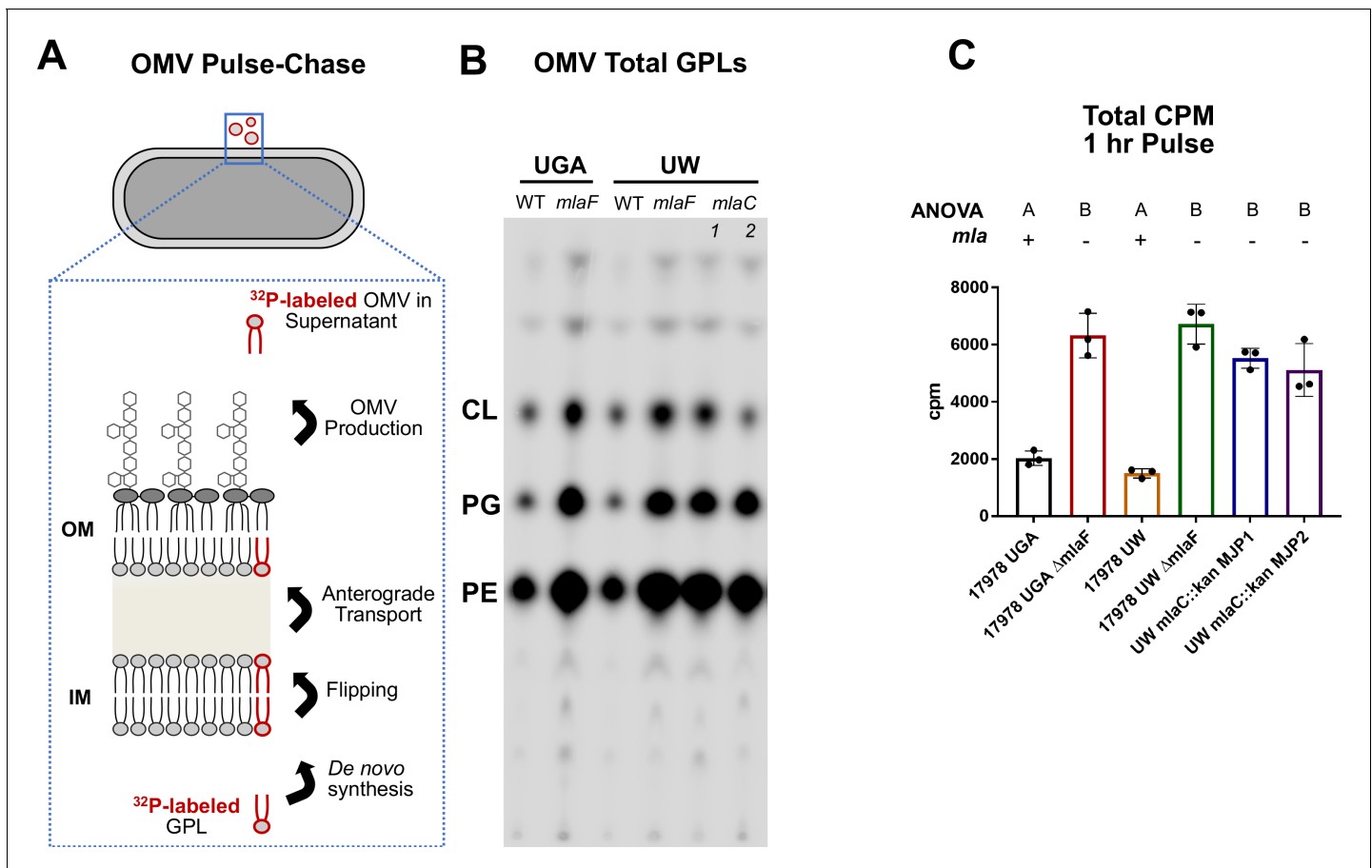

**Figure 2.** Mla mutants exhibit no defect in anterograde transport of GPLs. (**A**) Schematic of the biological principle behind the OMV pulse chase assay. Cultures were pulsed with $^{32}$P$_i$, which is incorporated in de novo synthesized lipids (red outlined GPL). After synthesis, lipids are flipped across the IM and anterograde transported to the OM. Over the course of the experiment, a percentage of lipids from the OM will be shed as OMVs. Any $^{32}$P-labeled OMVs must have incorporated GPLs that underwent anterograde transport. (**B**) Representative TLC of supernatant GPLs. Total sample was loaded after LSC quantification. GPLs were separated in solvent system containing chloroform, methanol, and acetic acid (65:25:10, v/v/v). (**C**) LSC quantification of extracted GPLs. Circles represent individual replicates. Lettering above denotes significantly different clusters as determined by a one-way ANOVA.

The online version of this article includes the following figure supplement(s) for figure 2:

**Figure supplement 1.** Membrane separations do not work in *A. baumannii*.
**Figure supplement 2.** Liquid scintillation counts of whole cells and OMVs from pulse-chase assay.

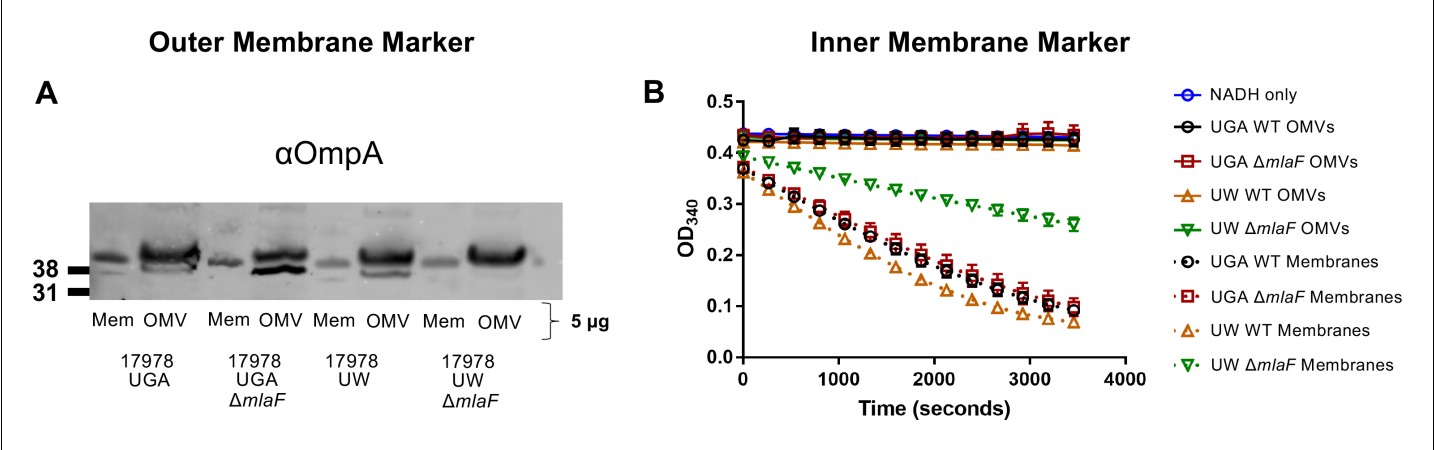

**Figure 3.** Validation of OMV pulse-chase assay. (**A**) α-OmpA blot of either OMVs or total membranes (Mem) for each genotype tested. Total protein was normalized to 5 µg. Blot is representative of biological duplicates. (**B**) NADH oxidase activity of OMVs or total membranes over time. For each sample, protein was normalized to 3 µg. Error bars are standard deviation of technical triplicates and graph is representative of biological duplicates.

When blotted for OmpA, we observed OmpA enrichment in the OMV samples relative to total membranes which served as our control. This is consistent with the fact that OMVs are exclusively OM material (*Figure 3A*). In contrast, when assessed for NADH oxidase activity, only total membranes were enzymatically active, confirming that OMVs contained no IM protein (*Figure 3B*). We will note that the UW *mlaF* total membranes do not exhibit as robust of NADH oxidase activity as WT, an additional demonstration of the aberrant physiology of this strain.

Furthermore, we expanded the pulse-chase quantification to total GPLs present in the cell pellet as there was the potential that de novo GPL biosynthesis was unexpectedly ablated over the 1 hr labeling period. This is not the case, as LSC quantification of GPLs extracted from these samples were consistent (*Figure 2—figure supplement 2*).

Lastly, an alternative interpretation of the OMV Pulse-Chase data that we would like to address is one in which anterograde transport was reduced with a concurrent increase in vesiculation. Assuming a 50% decrease in anterograde transport per Kamischke et al., OMV production would necessarily increase 200% to compensate and provide equivalent counts in the assay. Biologically, we find this scenario to be implausible due to the burden on the cell to maintain the OM. We also find this incompatible with the levels of OMVs harvested after 4 hr of growth (*Figure 1C*). In this case, we see relative degrees of vesiculation 2-4X higher than that of WT. From the pulse-chase assay, we see relative degrees of vesiculation within this same range, around 3-4X. That these ranges overlap strongly suggests that anterograde transport is unaffected in these strains. For Kamischke et al.'s finding to be masked by increased OM vesiculation, we would have had to seen a 6-8X increase in total OMVs independent of the Pulse-Chase assay, which we did not. Thus, our data strongly preclude the scenario as a possibility.

Altogether, this myriad of controls indicates that the OMV pulse-chase assay is a solid and robust approach for monitoring anterograde GPL transport.

## Transcriptomic variations between *mlaF* mutants highlight strain differences

With the conclusion that *mla* mutants do not exhibit any defect in anterograde transport, we turned our attention to why the UW Δ*mlaF* mutant showed phenotypic differences. We used RNA-seq to globally identify physiological changes that may be responsible. It was evident by principal components analysis (PCA) that the UW Δ*mlaF* strain had a significantly different transcriptomic profile compared to its isogenic parent (*Figure 4A*). By PCA, these profiles differed both in the PC1 and PC2 axes, which in total described 71.6% of the total variance amongst the datasets. In contrast, the UGA Δ*mlaF* mutant clustered much closer to its isogenic parent, consistent with results in another strain of *A. baumannii* published previously (*Powers and Trent, 2018a*). This was exemplified when we identified and compared significantly differentially expressed genes between the Δ*mlaF* mutants

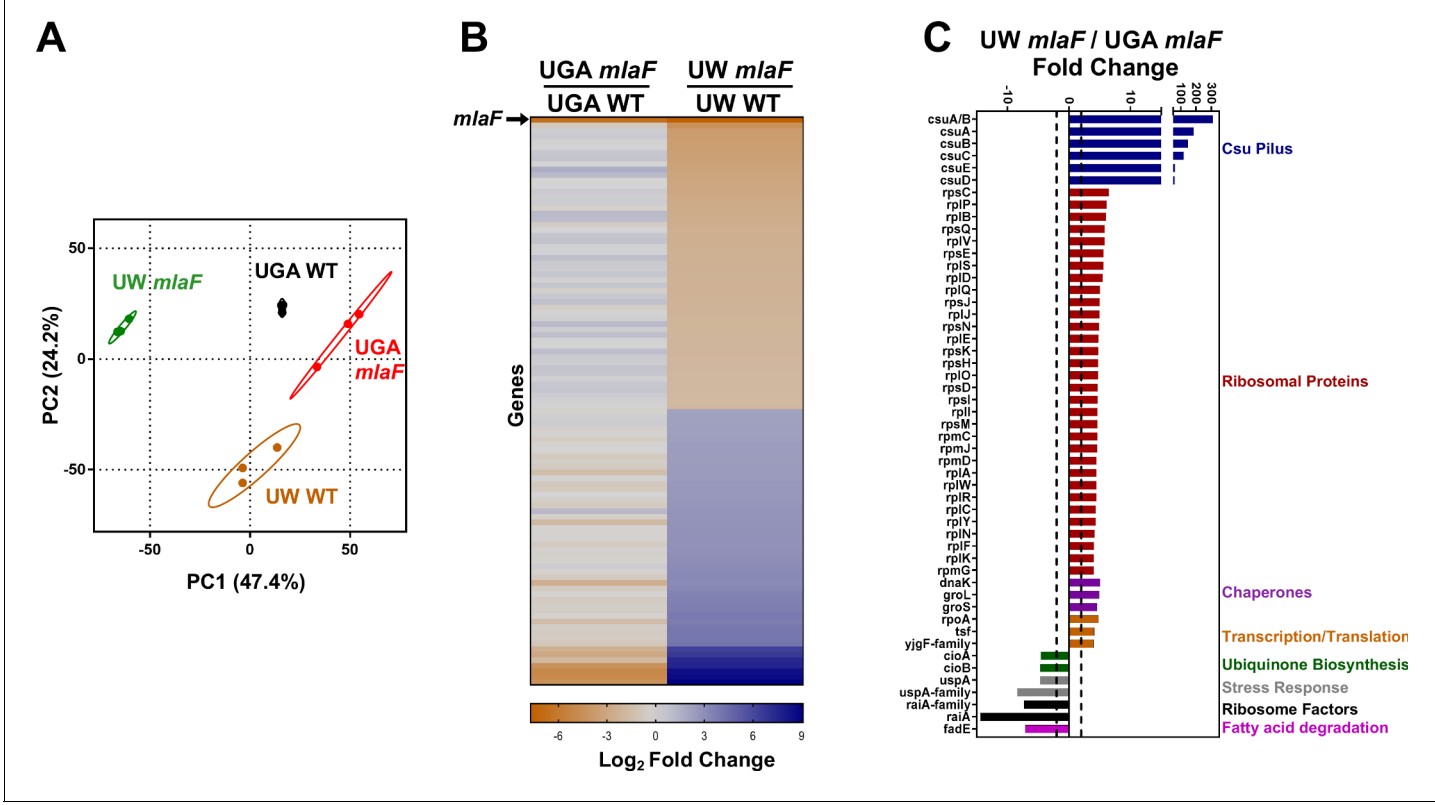

**Figure 4.** Transcriptomic profiling highlights substantial differences between ΔmlaF mutants. (**A**) Principal components analysis (PCA) of total transcriptomic profiles of UGA WT (purple square) and ΔmlaF (blue square) and UW WT (green circle) and ΔmlaF (red circle). Each point represents a biological replicate. (**B**) Heat map comparing differentially regulated genes between UW ΔmlaF and UW WT. All genes in the right column are differentially expressed with a > |2| log$_2$ fold change and a FDR p-value<0.05. For comparison, fold-change values of the same set of genes for the UGA ΔmlaF vs UGA WT are displayed in the left column. The row representing mlaF is denoted with an arrow. A full list of genes, fold-changes, and p-values for both comparisons are in **Supplementary files 1** and **2**. (**C**) Comparison of total transcriptomic profiles between UW ΔmlaF and UGA ΔmlaF mutants. Genes are clustered and colored by predicted or known function. Dashed lines denote +/- 2 fold-change. All transcriptomic data are presented as average RPKM values from three biological replicates.

and their isogenic parents. While the UGA ΔmlaF and WT had 13 genes that were differentially expressed excluding mlaF (**Supplementary file 1**), the UW ΔmlaF and WT differed by 103 genes (**Supplementary file 2**). When we looked at this array of 103 differentially regulated genes in the UGA ΔmlaF and WT, it was readily apparent that none of these changes were conserved between ΔmlaF mutants (**Figure 4B**). When we directly compared the UW and UGA ΔmlaF mutants, we identified multiple clusters of genes that were differentially regulated (**Figure 4C**). The first cluster are genes involved in Csu pilus formation, which were *significantly* upregulated in the UW ΔmlaF (**Figure 4C**). In fact, the *csu* operon was downregulated in the UGA ΔmlaF relative to WT (**Figure 4B**, **Supplementary file 1**). The Csu pili are involved in abiotic surface attachment; however, physiologically relevant functions remain underexplored (**Weber et al., 2016**). Importantly, Kamischke et al. reports an increase in biofilm formation in Δmla mutants in their study: this transcriptional data confirms that the UW ΔmlaF provided to us is behaving similarly in our hands (**Kamischke et al., 2019**). Interestingly, we found a large array of 70S ribosomal proteins were differentially regulated between the ΔmlaF mutants (**Figure 4C**). Additional genes involved in ribosome stability, transcription, and translation were differentially regulated (**Figure 4C**). That we observed global changes to ribosomal or ribosomal-adjacent proteins hinted at differences to ribosome dynamics, translation defects, and/or stringent response. To determine if there was a genetic basis to these observed transcriptional and phenotypic differences, we utilized whole-genome sequencing.

## Whole-genome sequencing identified an important ObgE variant

In Kamischke et al., three different mutants were used: ΔmlaF, ΔmlaC, and a dominant negative allele of mlaF expressed in trans. Based on the phenotypes reported, these strains behaved identically (*Kamischke et al., 2019*). We assumed that any mutation responsible for the phenotype of UW ΔmlaF must be present in the UW WT, as the statistical likelihood of a deleterious mutation accumulating independently in multiple strains was very low. Instead, if a mutation present in the UW WT was synthetically sick in conjunction with ΔmlaF, it could explain the phenotypes observed. We sequenced both the UGA and UW strains and looked for mutations unique to both the UW strains that were absent in the UGA strains. From this comparison, we were able to identify one non-synonymous mutation in the coding region of the essential gene obgE.

The obgE SNP in the UW WT and ΔmlaF resulted in a single amino acid change from an asparagine (N) to an isoleucine (I) at position 258 (N258I). While the N258 position is present in the reference genome, we wanted to confirm this specific residue was conserved amongst *A. baumannii* species. Comparison of sequences of obgE from 14 published genomes affirmed that the N258 position is highly conserved in *A. baumannii* (*Figure 6—figure supplement 1*).

## The UW ΔmlaF with ObgE* accumulates ppGpp

Due to the presence of the obgE* SNP in the UW strains, we sought to determine whether levels of stringent response nucleotides were altered as this would be the most obvious phenotype, particularly in the absence of mla. To that end, we labeled cultures with $^{32}$P$_i$ and extracted total nucleotides in formic acid. After separation by TLC, pppGpp, ppGpp, and GTP levels in both the UGA and UW strains were visualized and quantified via densitometry (*Figure 5* and *Figure 5—figure supplement 1*). What was immediately apparent was that the nucleotide profile of the UW ΔmlaF with obgE* was strikingly different from all other strains tested (*Figure 5* and *Figure 5—figure supplement 1*). The UW ΔmlaF was almost completely depleted of pppGpp, with a −12% shift when compared to WT (*Figure 5—figure supplement 1*). Concurrently, there were +3% and +9% shifts in ppGpp and GTP levels, respectively. In contrast, the UGA ΔmlaF strain maintained consistent levels of GTP relative to WT, with a −3% decrease in ppGpp and a +3% increase in pppGpp (*Figure 5—figure supplement 1*). This evidenced to us that the UW ΔmlaF strain exhibited abnormal stringent response alarmone profiles that were highly reproducible. It is important to note that relative levels of pppGpp, ppGpp, and GTP/GDP are interconnected (Figure 7). Increases in (p)ppGpp levels would increase downstream GTP/GDP levels as either species are generated by pyrophosphate hydrolysis of the alarmone: a scenario consistent with trends observed in our analysis. That we observed these increases in conjunction with a huge decrease in pppGpp strongly suggests to us that the nucleotide flux is directed toward ppGpp accumulation.

## ObgE* and Δmla are synthetically sick

Knowing that the UW ΔmlaF that harbors obgE* exhibited altered stringent response nucleotide profiles, we first wanted to confirm this phenotype in our hands by recreating ΔmlaF and ΔmlaC strains in the UW WT. Using the UW WT harboring obgE*, we generated deletions in either mlaC or mlaF using our recombination methodologies as previously described (*Powers and Trent, 2018a*; *Tucker et al., 2019*; *Tucker et al., 2014*). While we were able to generate mlaC and mlaF mutants, we found that they never exhibited the growth defect of the UW ΔmlaF mutant (*Figure 1—figure supplement 3A*). We were curious as to whether this had anything to do with the obgE* allele. Indeed, sanger sequencing revealed that in every ΔmlaC or ΔmlaF we generated, the obgE* SNP repaired itself: a phenomenon we observed 15/15 times (*Figure 1—figure supplement 3B*). Because of this, we were unable to replicate either the growth defect or the maintenance of the obgE* allele in any mla deletion.

An alternative approach was to reconstruct obgE* expressing strains de novo. This approach is complicated due to the need to first generate merodiploids as obgE is essential. We initially made either WT or ΔmlaC strains carrying a plasmid expressing obgE*. Afterwards, the chromosomal obgE locus was replaced with a kanamycin cassette, resulting in either obgE::kan, pMMB67EH-obgE* or ΔmlaC, obgE::kan, pMMB67EH-obgE*. As we expected expression of obgE* in conjunction with ΔmlaC would be deleterious, we used whole-genome sequencing to confirm a lack of suppressors in the constructed strains prior to in vivo analyses. Unfortunately, the ΔmlaC, obgE::kan,

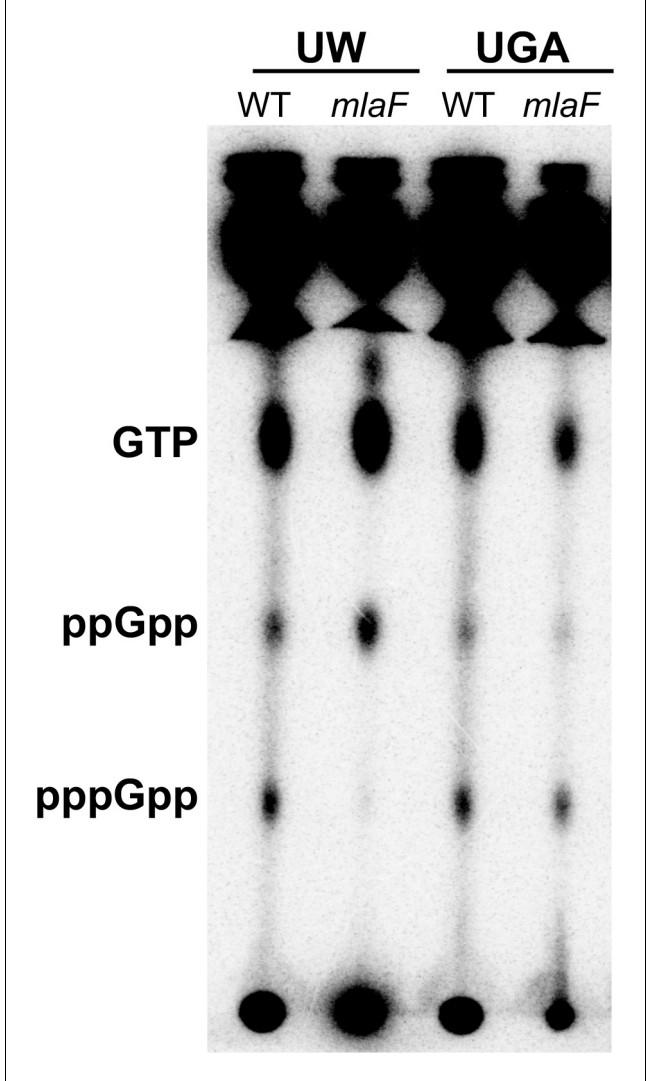

**Figure 5.** UW Δ*mlaF* mutant accumulates ppGpp.  (**A**) Representative TLC of $^{32}P_i$-labeled nucleotides extracted from whole cells separated on PEI-cellulose plates in 1.5 M $KH_2PO_4$ mobile phase. Positions of GTP, ppGpp, and pppGpp are denoted on the left. The image shown here is representative of three biological replicates. Additional replicate images are shown in *Figure 5—figure supplement 1*.

The online version of this article includes the following figure supplement(s) for figure 5:

**Figure supplement 1.** Nucleotide profile of UGA and UW strains.

pMMB67EH-*obgE** strains accumulated unique mutations with unknown implications that prevent legitimate phenotypic comparisons (*Supplementary file 3*).

Using two different methodologies, we were unable to generate a *mla*-null strain in the presence of *obgE**. Using standard recombination techniques, *mlaC::kan* and *mlaF::kan* strains repaired the native, chromosomal *obgE** mutation with a 100% frequency in our hands. Attempts to achieve the same result beginning with a merodiploid resulted in putative suppressors exclusively in the *mla*-null strains. Ideally, we would be able to replicate the exact phenotype of the UW Δ*mlaF* strain. In attempting to do so, we obtained genetic data that are supportive of our hypothesis that *obgE** and Δ*mla* have a synthetic phenotype. Even in the absence of a reconstructed strain, all major findings of this paper are directly derived from the UW strains provided to us.

ObgE is implicated in various physiological processes rooted in the regulation and homeostasis of the stringent response alarmone (p)ppGpp (*Persky et al., 2009*). Additional evidence shows ppGpp inhibits de novo fatty acid biosynthesis and by proxy GPL biosynthesis (*Heath et al., 1994*).

To understand the putative link between ObgE and Mla, we first must take into consideration critical phenotypes associated with Δ*mla* strains. These are three-fold. The first is that Mla-null mutants have compromised OM asymmetry (*Malinverni and Silhavy, 2009*). The second is that Mla-null mutants exhibit increased OM vesiculation (*Figure 1C*). Lastly, there is strong genetic evidence in *E. coli* that PldA-derived fatty acids stimulate de novo lipid A biosynthesis (*May and Silhavy, 2018*). Mislocalized GPLs are enzymatically degraded by PldA, resulting in fatty acids that increased lipid A biosynthesis downstream. Biologically, this would function as a feedback that asymmetry is perturbed and more LPS/LOS is needed in the outer leaflet.

One can reasonably assume that Mla-null mutants require increased GPL and lipid A biosynthesis to compensate for the loss of lipid through vesiculation occurring at the OM. If ObgE* perturbs stringent response homeostasis, one could hypothesize that in conjunction with Mla-null phenotypes, cells are incapable of sufficiently synthesizing lipids at a rate concurrent with lipid loss at the OM resulting in a synthetic sick phenotype (Figure 7).

Operating under this hypothesis, we should be able to probe the synthetic interaction between Δ*mla* and *obgE** using chemical biology approaches. For this, we used two compounds to target two different processes: serine hydroxamate to induce ppGpp accumulation and cerulenin to inhibit fatty acid biosynthesis. In either case, we would expect Δ*mla*, *obgE** strains to be hyper-sensitive to these compounds as either treatment should exacerbate the depletion in fatty acid pools. To test this, we treated cultures uniformly with either serine hydroxamate or cerulenin for 3 hr and compared them to untreated control cultures (*Figure 6*).

When we treated with serine hydroxamate, which induces stringent response, only the UW Δ*mlaF* strain was hypersusceptible with a 50% reduction in growth (*Figure 6A*). No other Δ*mla* strain tested exhibited a phenotype (*Figure 6A*). To confirm this treatment altered (p)ppGpp levels in the UW

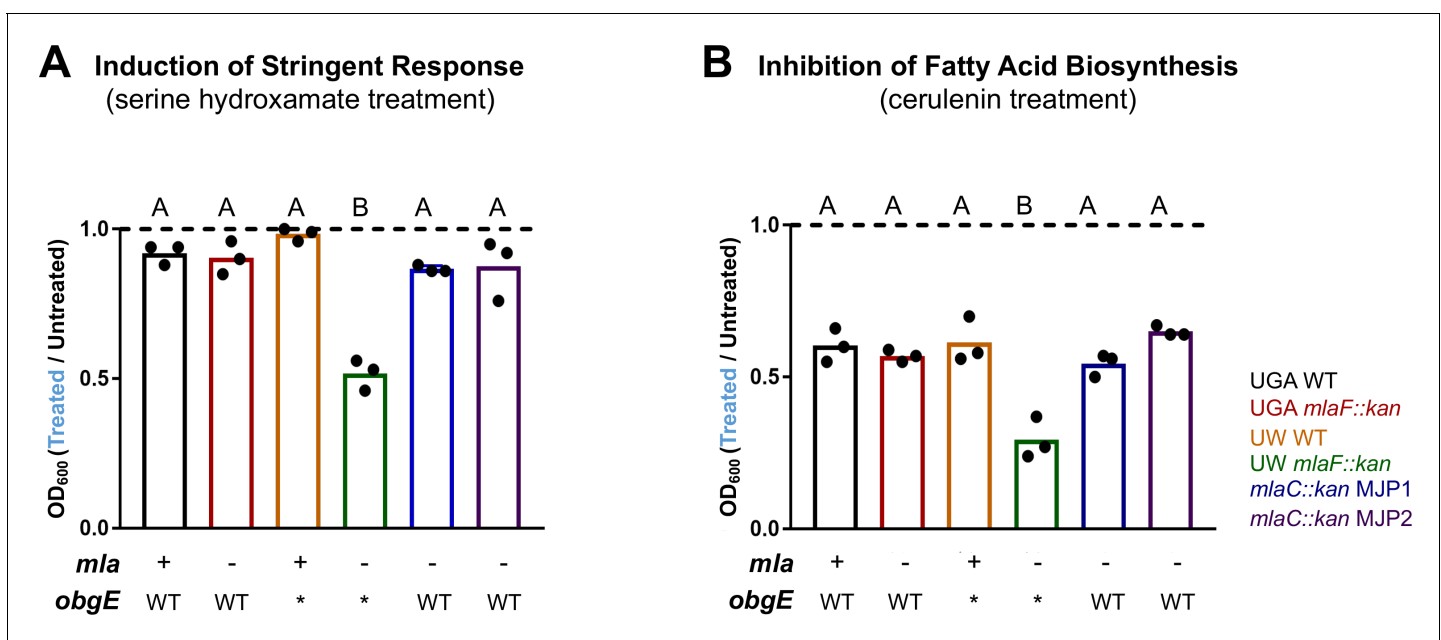

**Figure 6.** Δ*mla* with *obgE** is synthetically sensitive to chemical modulation of stringent response or fatty acid biosynthesis. (A) Normalized growth of cultures treated with 100 µg/mL serine hydroxamate which induces stringent response. Graphs represent normalized $OD_{600}$ readings of treated vs untreated. Dashed line at 1.0 indicates no difference in growth of treated culture. Values < 1.0 mean growth was decreased after treatment. Bars are colored by genotype with individual circles representing biological replicates. The *mla* and *obgE* alleles are indicated below for ease of comparison. Lettering above denotes significantly different clusters as determined by a one-way ANOVA. (B) Normalized growth of cultures treated with 100 µg/mL cerulenin which inhibits fatty acid biosynthesis. Graphs are displayed identically as panel (A).

The online version of this article includes the following figure supplement(s) for figure 6:

**Figure supplement 1.** Multiple sequence alignment of *obgE* across *A. baumannii*.
**Figure supplement 2.** Quantification of (p)ppGpp levels after serine hydroxamate treatment at 100 µg/mL.
**Figure supplement 3.** Increased stringent response inhibits de novo GPL biosynthesis.

Δ*mlaF* strain, we performed this assay in parallel with $^{32}$P-labeled cultures. After extraction and separation of nucleotides by TLC, we assessed relative levels of (p)ppGpp by densitometry. At this concentration of SHX, alterations to (p)ppGpp levels were relatively minor across strains, except for the UW Δ*mlaF,* which was even further exacerbated in it's accumulation of ppGpp (*Figure 6—figure supplement 2*). This corresponds with the UW Δ*mlaF* uniquely being sensitive to SHX at this concentration. Similarly, when we treated with cerulenin, which inhibits fatty acid biosynthesis, only the UW Δ*mlaF* strain was hypersusceptible (*Figure 6B*). In both cases, Δ*mlaC* deletions made in the UW WT (*mlaC::kan* MJP1 and MJP2) which had repaired the *obgE\** allele, looked identical to WT for both treatments (*Figure 6*).

To bolster confidence in the link between lipid biosynthesis and stringent response, we grew $^{32}$P-labeled cultures with or without serine hydroxamate at a higher concentration then isolated and quantified total GPLs. Total counts of the isolated lipid fractions were quantified by LSC and normalized by untreated controls per genotype. Across the board, de novo GPL levels were significantly reduced upon induction of stringent response with serine hydroxamate, confirming that elevated ppGpp levels indirectly inhibit GPL biosynthesis (*Figure 6—figure supplement 3*).

## Discussion

While the building blocks of the bacterial cell are synthesized in the cytoplasm, the cell envelope sets parameters for cell size and capacity (*Hughes et al., 2019*; *Ercan et al., 2019*). In Gram-negative bacteria, this is complicated further due to the presence of two membranes: the IM which separates the cytoplasm and the OM which protects the entire cell from the extracellular milieu. As such, mechanisms that mediate lipid transfer and homeostasis between and amongst membranes are of high importance to our understanding of Gram-negative physiology.

Here, we address a discrepancy in the role of one of these mechanisms: specifically, that of the Mla pathway. Mla, which has been confidently shown to be involved in lipid transport and homeostasis, remains controversial in the directionality that it operates (*Malinverni and Silhavy, 2009*; *Powers and Trent, 2018a*; *Kamischke et al., 2019*; *Hughes et al., 2019*; *Ercan et al., 2019*). We tackled this larger issue for the field to clarify work on the Mla pathway and lipid transport in Gram-negative bacteria more generally. This is critically important for both our basic biological understanding of bacteria but also in guiding development of new antibiotic targets and therapeutic treatments.

Due to the contradiction between our laboratory's results and the study published by the UW lab, we sought to examine further the role of Mla in *A. baumannii*. We began by obtaining an Δ*mlaF* mutant and its isogenic WT parent from the UW lab, which were graciously provided. While the UGA Lab Δ*mlaF* mutant exhibited no observable growth defects, the UW Lab Δ*mlaF* mutant grew significantly slower, exhibited smaller colony sizes on agar plates, and increased sensitivities to antibiotics. Furthermore, the UW Δ*mlaF* mutant exhibited a unique stationary phase lysis independent of its growth defect in exponential phase. These variables complicate the interpretation of the sensitive quantitative analyses performed in *Kamischke et al., 2019*. These phenotypes corresponded with a drastically altered transcriptomic profile. We further found that the UW Δ*mlaF* mutant decreased expression of *fadE,* the priming step in β-oxidation (*Campbell and Cronan, 2002*). We hypothesize this could help to redirect the flux of acyl-CoA pools toward elongation and away from degradation, which would help compensate for the loss of lipid from the OM.

While these data validated our hypothesis that the UW Δ*mlaF* strain was behaving uncharacteristically, it did not address the crux of the matter: is Mla in *A. baumannii* mediating anterograde transport? Despite our best efforts, we were unable to replicate robust, accurate membrane separations in *A. baumannii*. We obtained a more detailed membrane separation protocol from the UW Lab who had published successful membrane separations. Even with this protocol in hand, we were unsuccessful. Notably, another group independently verified that this gradient was insufficient to separate *A. baumannii* membranes (*Cian et al., 2020*). We will not speculate further as to why this was not reproducible, but it abolished our ability to test lipid transport using the assay described in Kamischke et al., as it relies entirely on membrane separations (*Kamischke et al., 2019*). Without being able to robustly separate membranes of *A. baumannii,* we were not confident in any degree of quantitative analysis with this methodology. To circumvent this issue, we devised a new assay that relies on the intrinsic nature of Gram-negative OM vesiculation. This OMV Pulse-Chase assay

exclusively monitors $^{32}$P-labeled lipids that were shed from living cells in a short period of time. As such, any radioactive lipids must have been de novo synthesized in the cytoplasm, *anterograde* transported to the OM, and ultimately shed into the supernatant. If Δ*mla* mutants were truly defective for anterograde transport to the degree reported in Kamischke et al., we would have expected to see approximately a 50% reduction in total counts in Δ*mla* mutants. We found the exact opposite. Regardless of the parent strain, every Δ*mla* mutant we tested in this assay exhibited an *increase* in radiolabeled OMVs. We find these data to be incontrovertible evidence that Δ*mla* mutants have no defects in anterograde transport. Indeed, the increase in radiolabeled OMVs is further evidence that Δ*mla* strains are defective for OM asymmetry and by proxy retrograde transport of mislocalized GPLs as initially proposed.

We remain confident in our conclusion that Mla in *A. baumannii* mediates retrograde GPL transport and our findings presented in our previous publication (*Powers and Trent, 2018a*). In efforts to prescribe a rationale for why the strains differed so drastically, we used whole-genome sequencing and identified a unique allele of *obgE* that mutated a conserved asparagine to isoleucine (N258I). It is unclear what this variant would do, which is further complicated by a minimal understanding of ObgE's exact biological function. However, we know that ObgE is linked to stringent response and ribosomal assembly. This was particularly interesting considering the RNA-seq, which highlighted multiple connections to the ribosome in the UW Δ*mlaF* mutant. Knowing this, we hypothesized that there must be a synthetic phenotype between *obgE** and *mla*. *A. baumannii* is tolerant of *obgE** under normal conditions; however, when membrane asymmetry is perturbed in Δ*mla*, its effects are exacerbated. We know that stringent response and ppGpp accumulation can have an array of downstream transcriptional effects including the inhibition of both de novo fatty acid biosynthesis and the incorporation of exogenous fatty acids (*Heath et al., 1994*). We argue this creates a negative feedback loop, where Mla-null cells bleb off lipids from the OM at a faster rate than the cell can synthesize de novo (*Figure 7*). This phenotype would be further exacerbated during the nutrient-deprived stationary phase. Indeed, we see significant stationary phase lysis of the UW Δ*mlaF* strain which harbors *obgE** further supporting this model.

Ultimately, we show multiple lines of evidence supporting this overarching hypothesis. Firstly, we present genetic evidence that *obgE** with Δ*mla* are actively selected against. In attempts to recreate Δ*mlaF* and Δ*mlaC* deletions in the UW WT (which natively has *obgE**), legitimate deletions repaired the *obgE** allele 100% of the time. These deletions lacked the growth defect of the UW Δ*mlaF* strain. Using an alternative approach, we generated *mlaC*$^+$ and Δ*mlaC* strains that exclusively expressed *obgE** from a plasmid with no native chromosomal copy. When we sequenced these strains to confirm their validity, the Δ*mlaC* strains had accumulated unique mutations that were not present in *mlaC*$^+$. We reasoned these were legitimate suppressor mutations that allowed for tolerance of *obgE** in the absence of Mla. While these are certainly of interest for fully teasing apart mechanism of *obgE**, we found further analysis beyond the scope of the current manuscript. Our focus here was to clarify for the field the role of the Mla system in lipid transport. Secondly, we used chemical biology to elucidate the effects of *obgE** in conjunction with Δ*mla*. If the overarching premise that these were incompatible due to fatty acid flux, we should be able to exacerbate the effects via the use of chemical inducers/inhibitors. Indeed, only cells with *obgE** and Δ*mla* were hyper-sensitive to increased stringent response/ppGpp levels and inhibition of fatty acid biosynthesis. When we looked at stringent response alarmone levels directly in vivo, we found that the UW Δ*mlaF* mutant accumulated significant levels of ppGpp at the expense of pppGpp, suggesting flux was directed towards ppGpp accumulation.

We argue that *obgE** in conjunction with Δ*mla* creates a synthetic sick phenotype linking fatty acid biosynthesis to OM homeostasis. Mla-null strains exhibit defects in lipid asymmetry of the OM, which result in increased OMV production. This blebbing of the OM would require compensatory de novo biosynthesis of both GPLs and lipid A, which both rely on de novo fatty acid biosynthesis. In this case, ppGpp accumulation would be simultaneously inhibiting de novo fatty acid biosynthesis: a circumstance that prevents cell envelope homeostasis (*Figure 7*). We conjecture that ObgE* could participate in one of two ways. The first would be stimulating hydrolysis of pppGpp to ppGpp (*Figure 7*, red arrow). While less likely, we cannot exclude the model that *obgE** could partially inhibit the terminal hydrolysis step of (p)ppGpp to GTP/GDP (*Figure 7*, red bars). At this point, either model is compatible with the data presented.

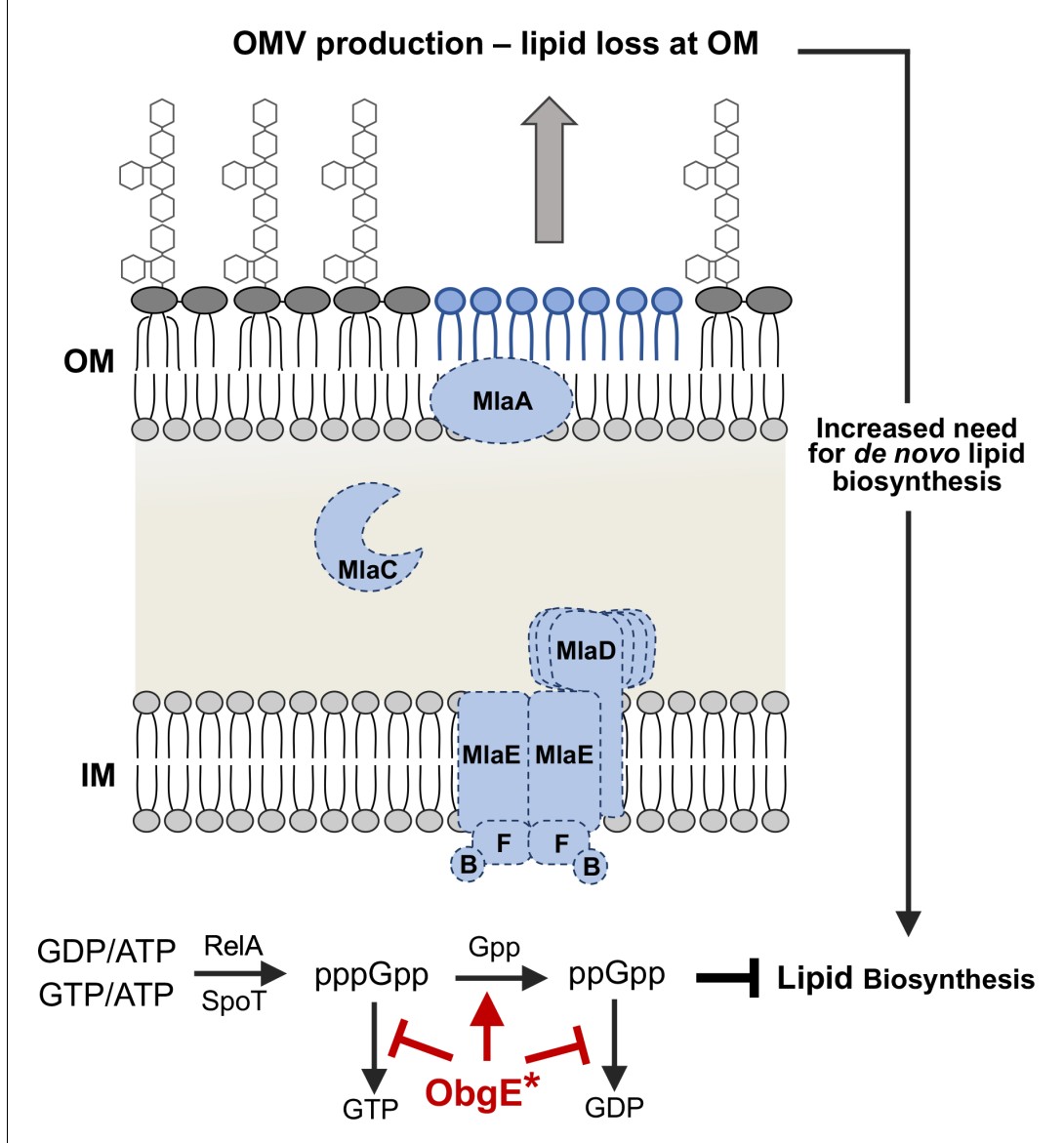

**Figure 7.** Model for the observed synthetic sick phenotype of Δ*mla* with *obgE\**. Mla mutants normally exhibit a 2-4X increase in OMV production, due in large part to the perturbed asymmetry of the OM (*Roier et al., 2016*). Additional evidence in *E. coli* suggests that fatty acids derived from OM phospholipase activity (PldA) stimulate lipid A biosynthesis (*May and Silhavy, 2018*). These factors result in an increased need for de novo fatty acid biosynthesis to contend with the rate of loss at the OM. Concurrently in the cytoplasm, the stringent response alarmone ppGpp has been shown to inhibit fatty acid biosynthesis (*Heath et al., 1994*). We know that Mla-null mutants (demonstrated here with dashed lines) with *obgE\** accumulate ppGpp at levels greater than that of WT (*Figure 5*, *Figure 5—figure supplement 1*). ObgE\* could be acting through either partial inhibition of the hydrolysis of (p)ppGpp to GTP/GDP (red bars) or stimulating the hydrolysis of pppGpp to ppGpp (red arrow). The resultant accumulation of ppGpp would repress fatty acid biosynthesis despite the global need for increased biosynthesis due to OMV production.

In this manuscript, we present data that strongly indicates Mla has no demonstrable role in anterograde transport in *A. baumannnii*. This unfortunately calls into question results presented in Kamischke et al. In our efforts to remedy the discrepancies between our papers, we identified a link between *obgE\** and the Mla pathway. We argue that the synthetic interaction between *obgE\** and Δ*mla* are responsible for the phenotypes observed in Kamischke et al. This synthetic phenotype is another link between cell envelope homeostasis and cellular stress response systems that broadly impact all aspects of bacterial physiology.

## Materials and methods

### Bacterial strains and growth curves

All strains and plasmids used in this study are listed in *Supplementary file 4*. Routine strain growth was done in lysogeny broth (LB) or LB + 1.5% agar at 37°C. Selective antibiotics were used when indicated at the following concentrations: kanamycin, 15 µg/mL (recombineering) or 30 µg/mL (routine growth); tetracycline, 10 µg/mL; hygromycin, 250 µg/mL. For growth curves, 5 mL of LB were inoculated with an overnight culture to a starting $OD_{600} \sim 0.05$. For a given time point, 200 µL of culture was transferred to a polystyrene 96-well plate. $OD_{600}$ readings were done using a H1 Hybrid Plate Reader (BioTek) and data was visualized in GraphPad Prism.

### Minimal inhibitory concentration determinations

For a given strain, an overnight culture was diluted 1:1000 in 18 mL of fresh LB. 4 mL of diluted culture were aliquoted to a fresh tube. Antibiotics were added at a final concentration of 800 µg/mL (vancomycin), 15.6 µg/mL (novobiocin), or 12.5 µg/mL (rifampicin). These samples were then 1:1 serially transferred using the initial diluted culture as diluent. Cultures were grown for 16 hr at 37°C and terminal $OD_{600}$ readings were measured. MIC determinations were calculated as the concentration at which 90% of growth was inhibited relative to an untreated control.

### Outer membrane vesicle quantification

An overnight culture was used to inoculate 250 mL of LB and grown for 24 hr. Cells were pelleted at 5000 x *g* for 10 min and the supernatant collected. This spin was repeated on the supernatant to remove any remaining cellular debris. Twice-spun supernatant was filtered through a 0.22 µm filter and stored at 4°C. Supernatant was centrifuged at 100,000 x *g* at 4°C using a Ti54 Beckman-Coulter rotor for a minimum of 3 hr. This process was repeated until all 250 mL of supernatant was pelleted. The resulting OMV pellet was washed once in 1X PBS and centrifuged overnight. The washed pellet was resuspended in 1X PBS. Quantification of OMVs used the colorimetric Purpald assay. Values were measured using an H1 Hybrid Plate Reader (BioTek) and converted to µg/mL of Kdo by using a standard curve generated with pure Kdo (Sigma).

### Next-generation sequencing

Cultures for RNA-sequencing were grown to an $OD_{600}$ of ~0.7. 1 mL of culture was pelleted and resuspended in 1 mL of RNAlater (Invitrogen). RNA extraction, library building, and sequencing were contracted with Genewiz, Inc Reads were mapped to the published ATCC17978 genome using CLC Genomic Workbench software (Qiagen). After local alignment, reads per kilobase of transcript per million reads (RPKM) was calculated for each coding sequence. A $\log_2$ fold-change of > |2| with an FDR p-value<0.05 were considered statistically significant. The PCA plot was generated using ClustVis (55). For whole-genome sequencing, cells were pelleted and gDNA extracted using the Easy gDNA Extraction Kit (Invitrogen). Libraries were built using the Nextera DNA Flex Kit (Illumina) per manufacturer instructions and sequenced on an iSeq100. Reads were trimmed, locally realigned, and variants mapped using CLC Genomic Workbench Software (Qiagen).

### Outer membrane vesicle pulse-chase

Fresh cultures were inoculated to an $OD_{600}$ ~0.05 from an overnight culture and grown at 37°C for 1 hr (2 hr in the case of the UW *mlaF* mutant to account for growth defect). $^{32}P_i$ was added to a final concentration of 10 µCi/mL and grown for an additional hour. Cultures were subsequently chased with 10 mM $K_2HPO_4$ for an hour. Cells were pelleted and supernatant carefully removed and transferred to a fresh tube. The supernatant was converted to a two-phase Bligh-Dyer via the addition of chloroform and methanol to a final ratio of 2:2:1.8 (chloroform:methanol:supernatant, v/v/v). For whole cell controls, pellets were initially resuspended in single phase 1:2:0.8 (chloroform:methanol:water, v/v/v). Insoluble material was removed by centrifugation and the supernatant was converted to a two-phase Bligh-Dyer. In both cases, glycerophospholipids were then extracted as described previously (*Powers et al., 2019*). Dried down lipids were resuspended in 500 µL of 4:1 chloroform:methanol (v/v) and 20 µL was quantified via liquid scintillation counting. The remaining sample was dried again under $N_2$ gas, resuspended in 20 µL of 4:1 chloroform:methanol (v/v), and the entirety

was spotted onto a Silica60 TLC plate. Glycerophospholipids were separated using a solvent system consisting of chloroform:methanol:acetic acid (65:25:10, v/v/v) and exposed to a PhosphorImager screen prior to visualization.

## Confirmation of OMV integrity

### Western blotting
5 µg of protein was loaded per sample and separated by SDS-PAGE on a 10% Bis-Tris gel. Protein was transferred to a 0.22 µm nitrocellulose membrane using a semi-dry transfer apparatus. The membrane was subsequently blocked overnight in 5% non-fat milk at 4°C. Primary antibody (α-OmpA) was used at a 1:5000 dilution and secondary antibody (α-Rabbit Cy5) at a 1:10,000 dilution. Blots were imaged for fluorescence using a Typhoon Imager (Amersham).

### NADH oxidase activity
NADH oxidation was monitored using absorbance as described below with a slight variation. Protein was normalized to 3 µg and $OD_{340}$ was monitored over time.

## Serine hydroxamate and cerulenin sensitivity
Growth curves were performed as described above with the addition of either 100 µg/mL serine hydroxamate or 100 µg/mL cerulenin. At the terminal time point, $OD_{600}$ values were normalized to the untreated controls.

## Radiolabeled nucleotide extraction
Fresh cultures were inoculated to an $OD_{600}$ ~0.05 from an overnight culture with the addition of 5 µCi/mL $^{32}P_i$. Cultures were grown at 37°C for 3 hr after which 500 µL were pelleted and resuspended in 100 µL of 2 N Formic Acid. Samples were incubated at RT for 15 min, frozen at −80°C for 15 min, and thawed at RT. This process was repeated an additional time. After thawing, samples were pelleted to remove cellular debris. 1 µL of supernatant was spotted on a PEI-Cellulose plate and separated in a 1.5 M $KH_2PO_4$ (pH 3.4) solvent system. Plates were exposed to a PhosphorImager screen for 2 days prior to imaging. Densitometry was performed in ImageJ. Densities for GTP, ppGpp, and pppGpp were calculated and relative percentages were derived for a given sample.

## Generation of mutants in *A. baumannii*
Standard mutant generation was performed using recombineering as described in *Tucker et al., 2019*.

## Sanger sequencing
For Sanger sequencing, PCR products were generated using Phusion polymerase. Products were purified with a PCR Purification Kit (Qiagen) and sent for sequencing at (Genewiz Inc).

## Membrane separations
An overnight culture was used to inoculate 250 mL of LB to an OD ~0.05. Cultures were grown to an OD ~0.8–1.0 and cells were pelleted at 5000 x g. Two protocols were used.

### Miller protocol
Pellet was resuspended in 12.5 mL 0.2 M Sucrose, 10 mM Tris (pH 7.5). While on ice, 18 mg of lysozyme was added and continuously stirred for 2 min. 12.5 mL of 1.5 mM EDTA was added and stirred for 7 min. This mixture was then centrifuged at 5000 x g and resuspended in 25 mL of 0.2 M Sucrose, 10 mM Tris (pH 7.5). Sample was lysed via high pressure homogenization at 10 kPsi (Constant Systems). Debris was removed via centrifugation at 7500 x g for 10 min. Total membranes were isolated via centrifugation at 4°C for 1 hr using a Ti54 (Beckman-Coulter). Membranes were resuspended in a small volume of 20% sucrose (w/v), 1 mM EDTA, 10 mM Tris (pH 7.5) and homogenized using a dounce homogenizer. This suspension was loaded onto a discontinuous gradient of 2 mL 73% sucrose (w/v), 4 mL 53% sucrose (w/v), and remainder 20% sucrose (w/v). All sucrose solutions contain 1 mM EDTA, 10 mM Tris (pH 7.5). Gradients were separated at 35,000 rpm in a SW41 (Beckman Coulter) for 16 hr.

## Osborn protocol

Pellet was resuspended in 8 mL of 10 mM Tris (pH 7.5), 0.75 M sucrose (w/v). While stirring on ice, 1 mg of lysozyme was added and stirred for 2 min. 16 mL of 1.5 mM EDTA (pH 8.0) was added dropwise over 5–10 min. The resulting slurry was sonicated for 30 s (1 s on/off) at 80% power (Qsonica). Sonication was repeated twice. Sample was centrifuged at 5000 $x$ $g$ and the resulting lysate was transferred to a fresh conical. Total membranes were isolated by centrifugation at 4°C at 100,000 $x$ $g$ for 1 hr in a Ti70 rotor (Beckman-Coulter). Membranes were dounce homogenized in 10 mM Tris (pH 7.5), 5 mM EDTA, 25% sucrose (w/w). This sample was layered onto a discontinuous gradient of 0.4 mL 60% sucrose (w/w), 0.9 mL 55% sucrose (w/w), 2.2 mL 50% sucrose (w/w), 2.2 mL 45% sucrose (w/w), 2.2 mL 40% sucrose (w/w), 1.3 mL 35% sucrose (w/w), 0.4 mL 30% sucrose (w/w). All sucrose solutions contain 5 mM EDTA, 10 mM Tris (pH 7.5). Gradients were separated at 35,000 rpm in a SW41 (Beckman Coulter) for 16 hr.

## Analysis of membrane markers

After centrifugation, gradients were carefully removed and fractions collected. In each case, a small hole was bored into the bottom of the tube and 500 µL fractions collected dropwise. Fractions were kept on ice.

## Protein concentration

50 µL of each fraction was measured for protein content via BCA (Thermo Fisher).

## NADH oxidase activity (inner membrane marker)

Fresh buffer containing 50 mM Tris (pH 7.5), 120 µM NADH, 0.1 mM DTT was prepared in ddH$_2$O. 5 µL (*E. coli*) or 10 µL (*A. baumannii*) of each fraction were added to 500 µL of buffer in a quartz cuvette. OD$_{340}$ was measured over 1 min. $\Delta$OD$_{340}$ and % activity was determined for each fraction across the entire gradient.

## LPS stain (outer membrane marker)

 SDS Loading buffer was added to 100 µL of fraction to a final concentration of 1X. 2 units of proteinase K was then added and samples incubated overnight at 55°C. 10 µL of sample were separated via SDS-PAGE using a 10% polyacrylamide gel. Staining for LPS was done using ProQ-Emerald 300 per manufacturer instructions (Invitrogen).

## Serine hydroxamate-treated lipid extraction

Freshly inoculated cultures (OD$_{600}$ ~0.05) were grown for 2 hr and pulsed with 10 µCi/mL $^{32}$P$_i$. Labeled cultures were immediately split in half and either exposed to serine hydroxamate (final concentration 1 mg/mL) or untreated. After an additional hour of growth, cultures were pelleted and glycerophospholipids extracted as described previously (*Powers et al., 2019*). Quantification of $^{32}$P-labeled lipids was done using LSC.

## Acknowledgements

We thank Sam Miller for graciously providing the WT and Δ*mlaF* mutant from their initial study as well as a detailed protocol of their membrane separations. This work was funded by NIH grants AI129940, AI138576, AI150098 (to MST) and FM137554 (BWS), and National Science Foundation Graduate Research Fellowship 049347–06 (to MJP).

## Additional information

### Funding

| Funder | Grant reference number | Author |
| --- | --- | --- |
| National Institute of Allergy and Infectious Diseases | AI129940 | M Stephen Trent |
| National Institute of Allergy | AI138576 | M Stephen Trent |

and Infectious Diseases

| National Institute of Allergy and Infectious Diseases | AI150098 | M Stephen Trent |
| National Science Foundation | 049347-06 | Matthew J Powers |
| National Institute of General Medical Sciences | 1F32GM137554-01 | Brent W Simpson |

The funders had no role in study design, data collection and interpretation, or the decision to submit the work for publication.

### Author contributions
Matthew J Powers, Conceptualization, Data curation, Formal analysis, Funding acquisition, Investigation, Methodology, Writing - original draft; Brent W Simpson, Conceptualization, Data curation, Formal analysis, Investigation, Methodology, Writing - review and editing; M Stephen Trent, Conceptualization, Supervision, Funding acquisition, Investigation, Methodology, Project administration, Writing - review and editing

### Author ORCIDs
Matthew J Powers ![ORCID] https://orcid.org/0000-0002-7896-3005
Brent W Simpson ![ORCID] https://orcid.org/0000-0002-1655-7407
M Stephen Trent ![ORCID] https://orcid.org/0000-0001-6134-1800

### Decision letter and Author response
Decision letter https://doi.org/10.7554/eLife.56571.sa1
Author response https://doi.org/10.7554/eLife.56571.sa2

## Additional files
### Supplementary files
- Supplementary file 1. Differentially regulated genes between UGA Δ*mlaF* and WT.
- Supplementary file 2. Differentially regulated genes between UW Δ*mlaF* and WT.
- Supplementary file 3. Unique mutations present in Δ*mlaC*, *obgE*::*kan*, pMMB67EH-*obgE*\*.
- Supplementary file 4. Strains and plasmids used in this study.
- Supplementary file 5. Primers used in this study.
- Transparent reporting form

### Data availability
Sequencing data (RNAseq) have been deposited to the database NCBI Gene Expression Omnibus. Accession number is GSE147139.

The following dataset was generated:

| Author(s) | Year | Dataset title | Dataset URL | Database and Identifier |
|---|---|---|---|---|
| Powers MJ, Simpson BW, Trent MS | 2020 | RNAseq of Mla mutants | https://www.ncbi.nlm.nih.gov/geo/query/acc.cgi?acc=GSE147139 | NCBI Gene Expression Omnibus, GSE147139 |

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
