## [Decision Letter]

**Acceptance summary:**

This study addresses the mysterious mechanism of lipid homeostasis in the bacterial Outer Membrane. A previous study published in *eLife* proposed that the Mla system promotes anterograde phospholipid transport from the IM to the OM. In this work, the authors provide compelling evidence that Mla is instead a retrograde transport system, promoting OM phospholipid turnover. In addition to previous works, the new evidence has potential to resolve the controversy and provide key clarifications into the function of this conserved system.

**Decision letter after peer review:**

Thank you for submitting your article "The Mla Pathway in Acinetobacter baumannii Does Not Mediate Anterograde Lipid Transport" for consideration by *eLife*. Your article has been reviewed by two peer reviewers, and the evaluation has been overseen by a Reviewing Editor and Gisela Storz as the Senior Editor. The reviewers have opted to remain anonymous.

The reviewers have discussed the reviews with one another and the Reviewing Editor has drafted this decision to help you prepare a revised submission.

The paper from Powers and co-workers investigates the role of the Mla system in glycero-phospholipid (GPL) transport in the Gram-negative bacterium Acinetobacter baumannii. The Mla system was discovered in *E. coli* just over a decade ago. Since then, significant evidence has been accumulated in the literature indicating that the Mla system is involved in maintaining lipid asymmetry in the outer membrane (OM) by promoting the retrograde transport of GPLs from the outer leaflet of the OM to the inner membrane (IM). However, last year in *eLife*, work from the Miller laboratory was published in which the authors argued that the Mla system in Acinetobacter is involved in anterograde transport of GPLs from the IM to the OM. By questioning the function of the Mla system, this work along with another biochemical/structural study have caused a controversy in the field.

The present report addresses this controversy by reinvestigating lipid transport in Acinetobacter strains defective for the Mla system. The authors obtained strains from the Miller lab used in the previous paper. They show convincingly that the *Δ-mlaF* strain in the Miller (UW) genetic background grows poorly and has major changes in its gene expression relative to an *mlaF* mutant in the Trent (UGA) background. The authors go on to show that these defects are likely due to a mutation in the essential gene obgE and that this mutation likely accounts in part for the defects in GPL transport reported by Miller and co-workers. In addition to oddities with the genetic background used by the Miller group, Powers and co-workers also present evidence that membrane fractionation to separate the IM and OM does not perform as expected in Acinetobacter. Relative to *E. coli*, the separation is quite poor. Effective separation of these layers is critical for assessing anterograde transport. The authors therefore devised a new assay for anterograde GPL transport that does not require membrane fractionation, and they find that inactivation of the Mla system does not reduce transport. The authors devise a new assay for anterograde GPL transport that does not require membrane fractionation, and they find that inactivation of the Mla system does not reduce transport.

The new assay however lacks appropriate controls (see below) and thus the results as currently presented in this paper only suggest, without truly establishing, that the Mla pathway does not operate in an anterograde direction. Another concern I have is that, although I think that Powers et al. may be right in their conclusions when they say that the Mla pathway is retrograde, their data do not completely rule out that Kamischke et al. could still be correct in the UW genetic background.

1) The new assay for anterograde GPL transport requires more investigation. The assay uses pulse-chase labeling of GPLs and then monitors their accumulation in OM vesicles (OMVs). Appearance of labeled GPLs in OMVs is a function of both anterograde transport from the IM to the OM as well as of the OMV formation process itself. Therefore, isn't it technically possible that slower anterograde transport in the mla mutants could be masked by enhanced (ie faster) OMV formation. For example, if anterograde transport were half as efficient but OMV formation were twice as fast, wouldn't the end result be the same as if both processes were normal? If this is the case, then the results of the assay are not as definitive as described in the text. Normalizations suggested above should be performed to more precisely quantify the amount of phospholipids present in the OM:

– Given that the UW *∆mlaF* mutant sheds more than UGA *∆mlaF* mutant (Figure 1B), that the quantifications in Figure 2C must be normalized against the amount of shedding; the authors should normalize their data by the total amount of lipid in the OMVs.

– The amount of LS counts contributed by cell lysis should be assessed, for example by measuring the levels of a cytoplasmic and of an IM protein in the supernatant after the pelleting of the cells and prior to lipid extraction. Also, to confirm that the OMVs they used were not contaminated by IMVs.

2) The authors explain that membrane separation is not reproducible in A. baumanii when two different techniques are used. How do the authors explain the differences between their work and Kamischke et al.? Could it be that membrane fractionation work better in the UW strain background than in the UGA? Kamischke et al. provided additional evidences in Figure 5—figure supplement 3 that their method of separation works actually pretty well. How come that it did not work for Powers et al? Did Powers et al. try the same methodology in the UW background? Could the discrepancy come from a difference in the physiological status of the cells? Kamischke et al. do not mention if cells were harvested in exponential or stationary phase. These points need to be discussed.

3) The negative association between the absence of the Mla pathway and a possibly overactive stringent response has not been studied in depth and is not supported enough by the data presented.

– If ppGpp accumulates in the *obgE** strain, wouldn't this be expected to decrease expression of ribosomal protein genes instead of increasing them as observed in Figure 3C?

– Discussion paragraph eight: How does hydrolysis of ppGpp yield GTP?

– Figure 4A: a quantification of ppGpp levels would be useful to support these results

The authors should have insisted more on this aspect as it offers an element of novelty.

4) TLC Experiments are semi quantitative raising potential issues with their interpretations:

– Figure 2B: a quantification of de novo GPL biosynthesis (as in Figure 6—figure supplement 3) would be needed to support these results

– Figures 5 and Figure 5—figure supplement 1: The data presented in these figures are not solid: the quantifications should have been performed using another method than TLC (semi-quantitative) followed by densitometry. Indeed, it seems that the total amount of nucleotides deposited on the TLC plate is not the same across conditions. An alternative strategy would be to separate the nucleotides, measure the LS counts associated to each of them, and normalize the nucleotide-specific counts against the total nucleotide count.

[Editors' note: further revisions were suggested prior to acceptance, as described below.]

Thank you for submitting your article "The Mla Pathway in Acinetobacter baumannii Does Not Mediate Anterograde Lipid Transport" for consideration by *eLife*. Your article has been reviewed by two peer reviewers, and the evaluation has been overseen by a Reviewing Editor and Gisela Storz as the Senior Editor. The reviewers have opted to remain anonymous.

The reviewers have discussed the reviews with one another and the Reviewing Editor has drafted this decision to help you prepare a revised submission.

The revised paper from Powers and co-workers is improved. The added controls and discussion of the results have addressed prior concerns related to the OMV assay. The only remaining issue with the paper is editorial. The text is repetitive and could be shortened. This paper is likely to be an important one in the field and therefore will have greater impact if its streamlined and easier to read. The paper currently devotes five paragraphs of the Introduction to summarize almost every aspect of the paper, which lacks concision. The "preview section" of the Introduction should be limited to one paragraph. This play-by-play summary is again repeated in the Discussion. While this is less of an issue here, it could also be paired down some to focus the Discussion on important concepts instead of dedicating a lot of text to repeating/summarizing the results.

---

## [Author Response]

The paper from Powers and co-workers investigates the role of the Mla system in glycero-phospholipid (GPL) transport in the Gram-negative bacterium Acinetobacter baumannii. […] . Another concern I have is that, although I think that Powers et al. may be right in their conclusions when they say that the Mla pathway is retrograde, their data do not completely rule out that Kamischke et al. could still be correct in the UW genetic background.

We acknowledge that this paper goes directly against a previous publication in *eLife*, and as such have done our best to address the reviewers’ comments. We believe we have addressed the concerns presented satisfactorily. We would like to emphasize that the major purpose of this manuscript is to rebut the evidence presented in Kamischke, et al. used to argue that Mla functions as an anterograde transporter. We believe this is of critical importance to the field and we hope the reviewers and the editor agree. Indeed, multiple preprints/publications have been released about Mla, much of it structural, that cite Kamischke, et al. as one of the major biological arguments for a function of anterograde transport. We believe it is in the best interest of the field that all relevant information on the Mla system is presented in the literature as soon as possible. Therefore, while we did our best to shore up the ppGpp/*obgE** connection, we would argue that any further mechanistic exploration of this relationship is beyond the scope of this article.

1) The new assay for anterograde GPL transport requires more investigation. The assay uses pulse-chase labeling of GPLs and then monitors their accumulation in OM vesicles (OMVs). Appearance of labeled GPLs in OMVs is a function of both anterograde transport from the IM to the OM as well as of the OMV formation process itself. Therefore, isn't it technically possible that slower anterograde transport in the mla mutants could be masked by enhanced (ie faster) OMV formation. For example, if anterograde transport were half as efficient but OMV formation were twice as fast, wouldn't the end result be the same as if both processes were normal? If this is the case, then the results of the assay are not as definitive as described in the text. Normalizations suggested above should be performed to more precisely quantify the amount of phospholipids present in the OM:– Given that the UW ∆mlaF mutant sheds more than UGA ∆mlaF mutant (Figure 1B), that the quantifications in Figure 2C must be normalized against the amount of shedding; the authors should normalize their data by the total amount of lipid in the OMVs.– The amount of LS counts contributed by cell lysis should be assessed, for example by measuring the levels of a cytoplasmic and of an IM protein in the supernatant after the pelleting of the cells and prior to lipid extraction. Also, to confirm that the OMVs they used were not contaminated by IMVs.

We agree with the reviewer that additional experimentation was needed here to control for aspects of the assay mentioned above. While theoretically, a 50% reduction in anterograde transport combined with a 100% increase in OMV formation would result in a net neutral effect, we find this scenario to be highly unlikely. We also find it incompatible with new data generated that shows the OMV vesiculation in the span of 4 hours to be within the same window of vesiculation from the pulse-chase assay. We have expanded upon this hypothetical scenario in the text and is included in subsection “Mla mutants exhibit no defects in anterograde transport”.

To validate the assay further, we included multiple controls which we hope allay the initial concerns about IM contamination. These data have now been reconfigured into Figure 1, Figure 1—figure supplement 2, Figure 2—figure supplement 2 and Figure 3.

Figure 1 has been rearranged to include cfu/mL data for 4 hours of growth (Figure 1B) and OMVs harvested after 4 hours (Figure 1C). This window of time mimics the maximum length of the pulse-chase assay. From the cfu/mL data, we know that increases in optical density during log growth are not due to lysis. Figure 1C shows that OMVs generated during logarithmic phase fall within the expected range for *mla* mutants.

These data indicate:

A) The time frame of the Pulse-Chase assay is within a time frame where we are not seeing exacerbated OMVs in the UW *mlaF* strain. Additionally, these values are not compatible with the hypothetical scenario of a 50% reduction in anterograde transport and a 200% increase in OMV generation.

B) Cell lysis is not occurring during the time frame of the Pulse-Chase assay.

C) The UW *mlaF* strain appears to exhibit significant stationary phase lysis, which would artificially inflate the Kdo quantification with cellular debris. We find this supports our proposed model of aberrant stringent response control in this strain. As these data are informative but not pertinent to the OMV Pulse-Chase assay, we have moved the stationary phase OMV data as a new supplemental figure, Figure 1—figure supplement 2.

Newly created Figure 3 contains controls for the presence of OM and IM proteins, respectively. Panel A is an αOmpA blot of either OMVs or total membrane controls for each genotype. As expected, OmpA is enriched in the OMVs relative to total membranes. Panel B is an enzymatic assay for NADH oxidase activity equivalent to that used for assessing integrity of membrane separations with one minor difference. In this case, total protein was normalized to 3 µg and NADH oxidase activity was monitored over time. The oxidation of NADH to NAD+ manifests in a decrease in absorbance of NADH at an OD_340._ None of the OMV samples exhibit any degree of activity, suggesting there is no contamination of IM in our harvested OMVs.

Taking these data into consideration, we are confident that our OMV assay is directly monitoring outer membrane vesiculation with no contamination by cell lysis nor by contaminating IM material over the maximum time that the assay takes place.

Text addressing these changes are now shown in Results subsections “Comparison of *mlaF* mutants” and “Mla mutants exhibit no defects in anterograde transport”.

2) The authors explain that membrane separation is not reproducible in A. baumanii when two different techniques are used. How do the authors explain the differences between their work and Kamischke et al.? Could it be that membrane fractionation work better in the UW strain background than in the UGA? Kamischke et al. provided additional evidences in Figure 5—figure supplement 3 that their method of separation works actually pretty well. How come that it did not work for Powers et al? Did Powers et al. try the same methodology in the UW background? Could the discrepancy come from a difference in the physiological status of the cells? Kamischke et al. do not mention if cells were harvested in exponential or stationary phase. These points need to be discussed.

We appreciate and share the reviewers’ concerns about how such disparate results were achieved by two separate groups. Indeed, if one looks objectively at Kamischke et al. Figure 5—figure supplement 3, particularly the images in panel F, it is clear they were able to obtain membrane separation. However, we will note several points.

Primarily, none of the membrane controls were done across the entire gradient, which is a point that was brought up several times per the published reviews of Kamischke et al. This is inappropriate for the technique used as membrane separation, even in *E. coli,* is never perfect. When membrane separations are not 100% perfect, assessing across the entire gradient ensures that you can consider every fraction that may contain your relevant membranes of interest. Kamischke et al. never reports these data, but rather shows a single OM and IM marker analysis for undescribed fractions. Without data across the gradient, it is simply impossible to determine the validity of membrane separations described. An image is not satisfactory, nor is it appropriate.

More concerning to us was a methods paper published in JoVE by one of the authors on the initial Kamischke et al. paper, Dr. Zachary Dalebroux. To be clear, this methods paper was NOT published prior to our submission to *eLife.* The title of this paper is “Separation of the Cell Envelope for Gram-negative Bacteria Into Inner and Outer Membrane Fractions With Technical Adjustments for Acinetobacter Baumannii.” The authors proceed to state:

“Recent work suggests that a derivation of the protocol we present here can be used to partition the bilayers of these organisms (Kamischke et al., 2019). Therefore, we tested our protocol on A. baumannii 17978. Initially, the procedure was inadequate.”

As such, it is not simply that we were unable to replicate the Kamischke et al. membrane separations. We now have an author of the original study that has published that the protocol used was not sufficient. We will state that we are remarkably confident in the results presented here and argue we have exceeded due diligence by utilizing multiple sucrose gradient methods and publishing marker analysis across the entirety of the gradient. We ask again for the reviewers to keep in mind the purpose of our current work is to inform the field and clarify the role of the Mla system in Acinetobacter baumannii.

We have chosen not to state in our manuscript that the recent JOVE methods paper was published by one of the co-authors of the original *eLife* article. The reader can determine that fact for themselves. The text has been adjusted as follows:

“Both our lab and the Dalebroux lab, have found the methodology to be insufficient as reported here and in JOVE (Cian et al., 2020).”

“Independently, a recent methods paper published during the review of this work demonstrated that the gradient utilized by Kamischke et al. was insufficient to separate *A. baumannii* membranes, further validating our results presented here (Cian et al., 2020)”

3) The negative association between the absence of the Mla pathway and a possibly overactive stringent response has not been studied in depth and is not supported enough by the data presented.– If ppGpp accumulates in the obgE* strain, wouldn't this be expected to decrease expression of ribosomal protein genes instead of increasing them as observed in Figure 3C?

It is difficult to speculate on this with the limited work done on stringent response in *A. baumannii.* This is further complicated by the fact that we are assessing the downstream consequences of an *obgE** mutation which could have numerous implications on how the stringent response is processed and maintained in this organism. ObgE has been shown to associate with the ribosome and plays a role in maintaining stringent response nucleotide ratios in *E. coli*. As such, one could imagine obgE* disrupts the ribosomal complex, and the cell responds by increasing ribosomal protein expression. There is certainly more work to be done to tease apart obgE* and stringent response in *A. baumannii.* However, here again we argue this is beyond the scope of the manuscript. Our goal is to address the discrepancy between recent work on the A. baumannii Mla system as it is impacting the cell envelope field as a whole.

– Discussion paragraph eight: How does hydrolysis of ppGpp yield GTP?

We apologize for this oversight which resulted in an oversimplification of the model. As the pyrophosphate bond is hydrolyzed, depending on the species pppGpp would result in GTP and ppGpp would result in GDP. We have updated the text and model figures accordingly.

– Figure 4A: a quantification of ppGpp levels would be useful to support these resultsThe authors should have insisted more on this aspect as it offers an element of novelty.

We have now included a quantification of (p)ppGpp levels that parallel the assay described in this figure, which are demonstrated in Figure 6—figure supplement 2. As you can see at this concentration of SHX, there are minor alterations to (p)ppGpp levels in all strains at this concentration of SHX with the exception of the UW *mlaF* mutant. The UW *mlaF* mutant (containing *obgE*)* exhibits a drastic alteration of nucleotide profiles, confirming its unique sensitivity at this concentration of SHX.

Text addressing these changes is in subsection “ObgE* and Δmla are synthetically sick”.

4) TLC Experiments are semi quantitative raising potential issues with their interpretations:– Figure 2B: a quantification of de novo GPL biosynthesis (as in Figure 6—figure supplement 3) would be needed to support these results

To further expand the OMV pulse chase assay, we included quantification of both GPLs and OMVs in two new replicates which are now displayed in Figure 2—figure supplement 2. As evidenced by these data, we are not witnessing major variations in GPL biosynthesis during the pulse-chase.

Text addressing this change is now present in subsection “Mla mutants exhibit no defects in anterograde transport”.

– Figures 5 and Figure 5—figure supplement 1:The data presented in these figures are not solid: the quantifications should have been performed using another method than TLC (semi-quantitative) followed by densitometry. Indeed, it seems that the total amount of nucleotides deposited on the TLC plate is not the same across conditions. An alternative strategy would be to separate the nucleotides, measure the LS counts associated to each of them, and normalize the nucleotide-specific counts against the total nucleotide count.

We agree with the reviewer that TLC densitometry is semi-quantitative and we do not argue otherwise. We followed a standard in the field, which uses nucleotide extractions coupled with TLC to assess relative levels of the nucleotides. While total amounts deposited on the plate may not be perfect, our densitometry assesses relative amounts. Arguably, by the quantification represented in Figure 5—figure supplement 1, the relative ratios of nucleotides are quite consistent. Even if you remove the semi-quantitative densitometry, the qualitative assessment of nucleotide distribution clearly shows that the UW *mlaF* mutant accumulates ppGpp in a reproducible fashion.

We are including several examples where TLC quantification of nucleotides was utilized in the literature:

https://www.pnas.org/content/115/29/E6845#s-1

https://jb.asm.org/content/202/12/e00045-20

https://www.ncbi.nlm.nih.gov/pmc/articles/PMC5845004/

https://www.ncbi.nlm.nih.gov/pmc/articles/PMC2771346/

[Editors' note: further revisions were suggested prior to acceptance, as described below.]

The revised paper from Powers and co-workers is improved. The added controls and discussion of the results have addressed prior concerns related to the OMV assay. The only remaining issue with the paper is editorial. The text is repetitive and could be shortened. This paper is likely to be an important one in the field and therefore will have greater impact if its streamlined and easier to read. The paper currently devotes five paragraphs of the Introduction to summarize almost every aspect of the paper, which lacks concision. The "preview section" of the Introduction should be limited to one paragraph. This play-by-play summary is again repeated in the Discussion. While this is less of an issue here, it could also be paired down some to focus the Discussion on important concepts instead of dedicating a lot of text to repeating/summarizing the results.

Thank you again for your consideration and review of this manuscript. We are excited for this work to be available for the field. We have done our best to streamline the Introduction.